# Biomedical Applications of the Biopolymer Poly(3-hydroxybutyrate-co-3-hydroxyvalerate) (PHBV): Drug Encapsulation and Scaffold Fabrication

**DOI:** 10.3390/ijms241411674

**Published:** 2023-07-19

**Authors:** Ana Isabel Rodríguez-Cendal, Iván Gómez-Seoane, Francisco Javier de Toro-Santos, Isaac Manuel Fuentes-Boquete, José Señarís-Rodríguez, Silvia María Díaz-Prado

**Affiliations:** 1Grupo de Investigación en Terapia Celular y Medicina Regenerativa, Instituto de Investigación Biomédica de A Coruña (INIBIC), Universidade de A Coruña, Complexo Hospitalario Universitario de A Coruña (CHUAC), Servizo Galego de Saúde (SERGAS), 15006 A Coruña, Spain; ana.rodriguezc@udc.es (A.I.R.-C.); javier.toro@udc.es (F.J.d.T.-S.); i.fuentes@udc.es (I.M.F.-B.); 2Grupo de Investigación en Terapia Celular y Medicina Regenerativa, Instituto de Investigación Biomédica de A Coruña (INIBIC), Complexo Hospitalario Universitario de A Coruña (CHUAC), Servizo Galego de Saúde (SERGAS), 15006 A Coruña, Spain; ivan.gomez.seoane@sergas.es (I.G.-S.); jsenrod@yahoo.es (J.S.-R.); 3Departamento de Fisioterapia, Medicina y Ciencias Biomédicas, Facultad de Ciencias de la Salud, Universidade da Coruña (UDC), 15006 A Coruña, Spain; 4Servicio de Reumatología, Complexo Hospitalario Universitario de A Coruña (CHUAC), Servizo Galego de Saúde (SERGAS), 15006 A Coruña, Spain; 5Centro de Investigacións Científicas Avanzadas (CICA), Universidade da Coruña (UDC), 15008 A Coruña, Spain; 6Centro de Investigación Biomédica en Red de Bioingeniería, Biomateriales y Nanomedicina (CIBER-BBN), 28029 Madrid, Spain; 7Servicio de Cirugía Ortopédica y Traumatología, Complexo Hospitalario Universitario de A Coruña (CHUAC), Servizo Galego de Saúde (SERGAS), 15006 A Coruña, Spain

**Keywords:** polyhydroxyalkanoates (PHA), Poly(3-hydroxybutyrate-co-3-hydroxyvalerate) (PHBV), encapsulation, scaffold, regeneration, drug delivery, nanoparticle

## Abstract

Poly(3-hydroxybutyrate-co-3-hydroxyvalerate) (PHBV) is a biodegradable and biocompatible biopolymer that has gained popularity in the field of biomedicine. This review provides an overview of recent advances and potential applications of PHBV, with special emphasis on drug encapsulation and scaffold construction. PHBV has shown to be a versatile platform for drug delivery, offering controlled release, enhanced therapeutic efficacy, and reduced side effects. The encapsulation of various drugs, such as anticancer agents, antibiotics, and anti-inflammatory drugs, in PHBV nanoparticles or microspheres has been extensively investigated, demonstrating enhanced drug stability, prolonged release kinetics, and increased bioavailability. Additionally, PHBV has been used as a scaffold material for tissue engineering applications, such as bone, cartilage, and skin regeneration. The incorporation of PHBV into scaffolds has been shown to improve mechanical properties, biocompatibility, and cellular interactions, making them suitable for tissue engineering constructs. This review highlights the potential of PHBV in drug encapsulation and scaffold fabrication, showing its promising role in advancing biomedical applications.

## 1. Introduction

In the last decades, the development and use of polymeric biomaterials regarding conventional polymers have increased in consideration. This is because these biomaterials have the advantage of being non-toxic, requiring less energy in their production, and are often obtained from waste from other industrial processes, used as raw materials for their creation. These properties make polymeric biomaterials ideal for improving and helping to protect the environment [1].

Biomaterials can be classified into two groups (which sometimes can be divided into three depending on the article being consulted): natural and synthetic biopolymers. The first one includes compounds such as collagen, fibrin, silk, and chitosan, but they also encompass biopolymers produced by bacterial fermentation, as is the case of polyhydroxyalkanoates (PHA). The second one includes biopolymers such as polylactic acid (PLA), polycaprolactone (PCL), and polyglycolic acid [2,3].

PHAs are a family of biopolymers discovered in 1926 by Lemoigne when he observed the bioaccumulation of these biomaterials in culture under conditions of the controlled fermentation of bacteria of the genus *Bacillus megaterium* [4]. It is currently known that these polymeric materials are produced by a wide variety of microorganisms when they are in an abundant carbon source and lack other nutrients such as nitrogen, phosphorus, and oxygen [5]. PHAs are synthesized as water-insoluble granules in the cytoplasm of cells. These biopolymers play a crucial role in helping microorganisms survive in challenging stress conditions [6].

This group of biopolymers can be formed by between 600 and 35,000 (R)-hydroxy fatty acid monomeric units [7]. The structure and some typical polymers are shown in Figure 1. More than 150 PHA monomers have been identified, making it possible to synthesize many materials where properties can be modified as needed [8]. There are two ways to classify monomers. The first one is based on the number of carbons they contain, in which case they can be categorized into three groups: short-chain-length PHAs (SCL, 3–5 carbon atoms), medium-chain-length PHAs (MCL, 6–14 carbon atoms), and long-chain-length PHAs (LCL, 15 or more carbon atoms) [7]. They can also be classified based on whether they are homopolymers or copolymers. Homopolymers are composed of a single type of monomer, such as poly (3-hydroxybutyrate) (P3HB), while copolymers are composed of two or more types of monomers, as is the case with poly(3-hydroxybutyrate-co-3-hydroxyvalerate) (PHBV) [9].

The specific properties of a copolymer are determined by the quantities of their respective monomers. An example is PHBV, whose mechanical and thermal properties can be controlled by adjusting the concentration of 3-Hydroxyvalerate (3HV) units in the copolymer. When the monomer content of 3HV is higher, it results in lower crystallinity, leading to increased flexibility, strength, and elongation at break [10].

Conventional polymers have the drawback of persisting in the environment for extended periods and being resistant to degradation by microorganisms. These materials accumulate in ecosystems, producing numerous toxic compounds. One of the reasons why they are still used in many fields is due to their lower cost compared to PHAs. Hence, novel strategies related to the fermentation and extraction of these biopolymers are being researched and improved [11].

A novel PHA synthesis process that is much cheaper than the conventional one, where pure cultures and synthetic substrates are used, is being studied. It is mainly based on the use of substrates from waste or industrial by-products, such as whey. The production process includes a first stage to produce volatile fatty acids and a second stage to produce the biopolymer. Through this process, in addition to solving an environmental problem, waste is valued [12].

A brief outline of developments related to PHAs is given in Figure 2 [13,14].

### 1.1. Properties of Polyhydroxyalkanoates

In the field of biomedicine, materials must have adequate physical-chemical and biological characteristics to be biocompatible, biodegradable, and non-toxic to humans.

#### 1.1.1. Biodegradability

In nature, PHAs can be broken down by the hydrolytic activity of the depolymerase secreted by some microorganisms. This process results in the breakdown of biopolymers into monomers and oligomers that can be used as nutrients by microorganisms. Once metabolized, they are converted into water and carbon dioxide [15]. This process is shown in Figure 3. The rate of this activity depends on some factors like the crystallinity, the composition, the surface area of the polymer, and various environmental conditions, including temperature, pH, and degradation rate. Additionally, the monomeric composition affects the rate of degradation. Polymers containing only 4-hydroxybutyrate monomer units degrade more rapidly than P(3HB) or PHBV copolymers [16].

#### 1.1.2. Biocompatibility

As mentioned above, PHAs are polymers synthesized as granules in the cytoplasm of microorganisms such as bacteria, which can also degrade the material itself for reuse as nutrients. This natural origin suggests a certain compatibility with biological systems. In addition, in this article, cell viability, adhesion, and proliferation in the presence of PHAs have been demonstrated with different cell lines. The results show that they can develop and work normally in contact with biopolymers.

One example of biocompatibility with the human body is the case of P(3HB); once degraded by enzymatic hydrolysis it forms 3-hydroxybutyric acid, a constituent belonging to blood plasma. Also, this metabolite is present in the brain, heart, lungs, kidneys, and liver. In the medical field, it has been used as an anesthetic, for the treatment of alcohol dependence and narcolepsy [18].

#### 1.1.3. Toxicity

Despite being biocompatible, it is very important that they do not cause toxicity to the host. To solve this problem, once the polymer is extracted, it is crucial to perform exhaustive purification to eliminate all the impurities that may remain attached to the biopolymer. This is very important, especially in the case of PHAs from Gram-negative bacteria, since they have a lipopolysaccharide layer on their outer membrane with a large toxic effect [19]. These endotoxins can be eliminated with some chemical treatments using sodium hypochlorite, sodium hydroxide, hydrogen peroxide, and benzoyl peroxide. For example, PHBV has been reported to cause inflammatory reactions in mice due to the impurities and components present in the biopolymer [18].

### 1.2. Poly(3-hydroxybutyrate-co-3-hydroxyvalerate)

PHBV is a copolymer originated from the incorporation of (3HV) monomers in P(3HB) biopolymer. The structure is shown in Figure 4. The incorporation of these monomers makes P(3HB) less fragile, which is the principal limitation to replace polypropylene, making it more flexible and resistant [20]. The amount of (3HV) in the polymer also affects its structure, as higher concentrations result in a more amorphous particle. Note that the release of the encapsulated drug or factor is not based on matrix degradation, but on diffusion through the amorphous regions of the sphere, so its surface modification plays a very important role in its biodegradability [21].

In addition, PHBV has high immunotolerance, greater chemical inactivity, and has already been developed on an industrial scale. Some limitations of this material include its marked hydrophobicity, high fragility, low impact resistance, and poor thermal stability. However, these problems can be addressed by reinforcing PHBV [22] with other materials, such as other polymers, natural fibers [23], nanometals [24], nano-cellulose [25], and carbon nanotubes [26,27] among others. A concrete example is the use of bacterial cellulose (BC) which is a sustainable polysaccharide produced by bacteria that can be mixed with PHBV. Toxicological studies have shown that BC is non-toxic and does not provoke inflammatory responses or oxidative stress at the cellular level. In vitro and in vivo tests confirm the safety of this biopolymer in medical applications. Both BC and PHBV meet the requirements for clinical use [28].

## 2. Applications

The mentioned properties of PHAs have allowed these polymers to be used in the field of biomedicine in different ways. Some examples include their use as sutures and prostheses, pericardial patches, scaffolds for tissue regeneration, and for controlled drug or compound release (Figure 5) [18].

### 2.1. PHBV Composites for Drug Delivery Applications

Drug encapsulation and delivery systems provide significant pharmacological advantages, such as enhancing the physical and chemical stability of encapsulated active ingredients, increasing bioavailability, acting as controlled-release systems, reducing fluctuations in drug concentrations in the blood, penetrating specific barriers and tissues, reducing adverse effects, and protecting the encapsulated molecule [29].

To achieve better control over drug release, some strategies have been developed to vary particle size distributions and degradability. One important fact is that pore formation must be avoided since it promotes rapid drug release. The control of particle size and porosity is achieved by optimizing parameters such as solution concentration, surfactant concentration, the polymer molecular weight, the homogenization technique, solvent characteristics, and the environment [30]. In the case of PHBV, studies have shown that the compound’s molecular weight and biomaterial concentration are related to an effect on particle size [31]. In other articles, the selected formation technique determines the size and shape of the nanoparticles. If they are produced by emulsification/evaporation, the nanospheres are round and porous, with a very wide size distribution, while for nanoprecipitation, which is a simpler and cheaper process, the particles are smaller [32].

Due to their characteristics, PHAs have been used for the synthesis of microspheres, microcapsules, or nanoparticles for the treatment of various diseases.

Regarding liver cancer, PHBV nanoparticles could treat one of the leading causes of death, hepatocellular carcinoma (HCC), typically caused by chronic infections such as hepatitis B and C. One study proposed a solution based on using a drug delivery system loaded with paclitaxel (PTX) and coated with pH-sensitive dopamine, which helps achieve controlled drug release. Additionally, RGD proteins (HCC-targeted arginine-glycine-aspartic acid peptide) were added to the nanoparticles, which can recognize other specific proteins on the cancer cell surface. Results demonstrated good anti-cancer efficacy and lower toxicity in comparison to free drugs in an in vitro and in vivo hepatocyte experiments [33].

Regarding colon cancer, PHBV and poly (lactic-co-glycolic acid) (PLGA) nanoparticles with encapsulated 5-Fluorouracil (5-FU) were developed. It was observed that this common drug used for this disease was less toxic when encapsulated. Furthermore, the drug derived from the nanoparticles showed better results in reducing tumor volume than the drug in the free form [34]. In the previous year, the same research group investigated the release of two drugs, the mentioned 5-FU and oxaliplatin, using the same nanoparticles. The results showed that they are a good option for intravenous administration because of their hemocompatibility [35]. A different research group conducted characterization and biocompatibility tests on PHBV nanoparticles, also using 5-FU for the same purpose. The size and morphology were studied using scanning electron microscopy (SEM). Cytotoxicity was evaluated using human adenocarcinoma cells. The results showed a significant reduction in cell viability, indicating that drug-loaded nanoparticles were a promising method for killing cancer cells [36]. In the past, the same group used PHAs nanoparticles to encapsulate Silymarin, a poorly water-soluble drug. The in vitro results showed its ability to reduce viability in colon cancer cells (HT-29). The nanocarriers also demonstrated the ability to penetrate 3D microtumors and reduce their size [37].

When it comes to breast cancer, the drug Docetaxel is commonly used. The principal limitation is that it is partially eliminated by the liver and kidneys when administrated via systemic circulation. This loss causes an increment in dosage that may lead to side effects. A solution is to use PHBV nanoparticles as a method of controlled drug release. Results showed that docetaxel-loaded nanoparticles could protect the drug in the biological fluid and increase cytotoxicity in vitro by inducing apoptosis in breast cancer cell lines (MCF7). In vivo studies have shown even better results due to the synergistic effects of enhanced permeation and retention and longer plasma circulation time [38,39].

The behavior of PHBV nanoparticles was also investigated in two different epithelial cell lines: HeLa (cervical cancer cells) and SKOV-3 (ovarian cancer cells). The results showed that the mechanism of absorption of PHBV nanospheres depended on time, energy, concentration, and the internalization mechanism of the cell line. The destination of the nanoparticles was also determined. In both cell lines, the nanoparticles ended up in the lysosomes where they were degraded, releasing their content (in this case markers). The study provided a novel insight into the internalization of PHBV nanoparticles [40]. In a similar line, another research group studied nanoparticles loaded with the mentioned PTX, and they found that it caused significant cellular death in primary cultures of stage III ovarian cancer cells from six patients [41].

Against the human lung cancer cell line A549, PHBV nanoparticles loaded with the drug ellipticine demonstrated significant anticancer activity [42]. The nanoparticle was synthesized in the past by the same group, using *Bacillus cereus* FB11, under nutritional stress conditions and using glucose as the only carbon source [43]. Another group investigated the cytotoxicity of nanoparticles loaded with sunitinib for the same purpose. The main objective, in this case, was to create an inhalable powder containing the nanoparticles with the encapsulated drug to increase the selectivity of lung cancer treatment. The article explains that in vivo studies are needed to confirm the therapeutic potential (Figure 6) [44].

Some studies used folic acid-functionalized PHBV nanocarriers loaded with the drug etoposide for the treatment of osteosarcoma. The 5-year survival rate for patients with localized osteosarcoma treated only with surgery went from 15 to 20%. When combined with chemotherapy agents like etoposide, this rate increased to approximately 70%. Researchers in this field performed the characterization of these functionalized PHBV nanocarriers using SEM to examine the morphology of the nanoparticles, Fourier Transformed Infra-Red spectroscopy (FTIR) for the structural characterization, and dynamic light scattering for all the sizes and size distribution parameters. The efficacy of encapsulation and drug release were also studied. The cytotoxic, apoptotic, and necrotic effects of PHBV nanoparticles were also investigated using Saos-2 osteosarcoma cells. The results showed that the cytotoxicity effect efficacy of etoposide-targeted loaded PHBV nanoparticles was higher compared to etoposide-untargeted loaded PHBV nanoparticles, and only etoposide. They also found that the cytotoxicity was significantly affected by the FA conjugation. In conclusion, the material was suitable for targeted drug treatment [45].

Nanoparticles have another application in the field of photodynamic therapy (PDT) for various types of cancer. In this technique, a photosensitizer (PS) is encapsulated in the nanoparticles, which can accumulate in cancerous tissues. When it is excited with a specific wavelength of light in the presence of molecular oxygen, the PS produces radical oxygen species that locally kill cancer cells. A research group used a nano-precipitation method to prepare a PHBV nanocarrier stabilizer with polyethylene glycol (PEG) lipids. These studies demonstrated the system’s efficacy for PS administration (Figure 7) [46]. Another research developed and characterized nanoparticles made with the same material, but in this case, functionalized with super-magnetic iron oxide nanoparticles (SPION) and loaded with the ceftiofur antibiotic. The article highlights the potential of this method to guide, concentrate, and release the active drug at a specific location [47]. Studies conducted previously, only with the SPION encapsulated in PHBV, have shown to be useful as contrast agents for magnetic resonance imaging and to have the potential for site-specific drug transport [48].

Another function of nanoparticles is to reduce the frequency of injections in treatments such as osteoporosis, a disease that makes bones weaker and causes severe and painful fractures in patients, mainly older people. As a solution, one group analyzed PHBV/PLGA nanoparticles as a controlled delivery system [49]. One year later, they prepared a combined delivery system using the PHBV/PLGA nanoparticles in hyaluronic acid and jeffamine hydrogel loaded with the drug teriparatide. Cytotoxicity and in vivo studies in mice were conducted. The results indicated that this combined delivery system was a promising and effective candidate for the treatment of osteoporosis [50].

PHB-PEG and PHBV-PEG nanoparticles loaded with epirubicin were studied to discover their antibacterial properties. The results showed that drug release was sustained in time for 8 days at pH 4 (more than at pH 7). The nanoparticles exhibited significant antibacterial properties against strains of *Staphylococcus aureus, Escherichia coli,* and *Pseudomonas aeruginosa*, compared to the free drug. [51]. In another group, researchers used plant leaf extracts such as *Psidium guajava* (guava)*, Raphanus sativus* (radish), *Solanum pseudocapsicum* (winter cherry), *Mentha royaleana* (mint), and *Calotropis procera* (rubber tree) as antibacterial compounds. PHBV nanoparticles proved to be efficient carriers and produced significant inhibition zones against multi-drug-resistant bacterial strains [52]. Other authors, using a high-speed double-emulsion technique, obtained PHBV nanoparticles loaded with quercetin, an antimicrobial, anti-inflammatory, and antiviral polyphenol. The results suggested that the encapsulation efficiency was 51%, they were non-cytotoxic, and the compound release occurred within the first 5 h of immersion in water [53].

Fingolimod is a synthetic compound that suppresses the immune system by the activation of certain receptors in lymphocytes. This drug has also been encapsulated in PHBV nanoparticles, and it was observed that its release showed an initial burst release of 24% in the first 24 h, followed by a sustained release over the next 4 weeks [54].

A very striking utility of nanoparticles is their use in the treatment of diabetes. One group encapsulated insulin in PHBV nanocarriers to create a sustained drug-delivery system that can maintain blood glucose levels within a normal range. The results demonstrated a pattern of prolonged drug release for up to 27 days [55].

PHBV nanoparticles have also been tested as a topical ophthalmic drug delivery system. The nanocarriers were loaded with hydrocortisone using an emulsification/solvent-evaporation technique. In addition to nanoparticle characterization, drug release, penetration of the nanocarriers into corneal tissues, and ocular cytotoxicity in bovine keratocytes were also investigated. The results suggested that the delivery system could be a good option for topical ophthalmic drug delivery [56].

PHBV nanoparticles also have potential in the treatment of Parkinson’s disease. In a study, these nanocarriers were used to deliver the drug pramipexole. The objective was to reduce the drug’s plasma fluctuations by subcutaneous injection of nanoparticles. The encapsulation efficiency and drug loading were determined using ultraviolet–visible spectrophotometry. The release profile was evaluated using a dialysis method. The results demonstrated that the drug-delivery system exhibited high encapsulation efficiency and drug loading, in addition to a sustained release profile [57].

Another potential application for nanocarriers is as transdermal therapy for skin conditions such as psoriasis, ultraviolet damage, and aging. In an article, the nanospheres were prepared using the oil-in-water (o/w) technique. Cytotoxicity of the particles was assessed using the 3-(4,5-dimethylthiazol-2-yl)-2,5-diphenyltetrazolium bromide assay (MTT) with L929 mouse fibroblasts, BALB/3T3 mouse embryo fibroblasts, and human HaCaT keratinocytes. The genotoxicity of the carriers was determined using the Ames test. The ability of the particles to penetrate the membrane of human osteosarcoma Saos 2 cells and accumulate in their cytoplasm was also evaluated. Additionally, penetration through human skin was investigated using samples obtained from a 42-year-old female donor after undergoing breast surgery. All tests had satisfactory results, indicating that the nanoparticles can be used for personalized treatment through topical administration [58]. In another study, nanoparticles with different sizes and the same material containing Nile red were employed to investigate the penetration of nanocapsules into healthy mouse skin. The penetration was assessed using gas chromatography–mass spectrometry analysis. The results revealed that smaller particles exhibited higher efficacy in penetrating the skin, and the size also played a role in controlling the depth of penetration. Moreover, lower concentrations of nanoparticles were found to be more effective [59].

The main encapsulation systems used in the articles mentioned above were double emulsion and nanoprecipitation. The first one provides excellent encapsulation efficiency and control over the drug-release profile, while the second one is advantageous due to its easy application in large-scale systems. It is important to note that many of these techniques produce residual organic solvents, such as chloroform for PHA extraction and methanol for PHA precipitation, which could cause a health risk if not completely removed prior to use in an in vivo study.

Regarding the materials, many authors choose to use blends that allow them to improve the properties of PHBV. One of the most commonly used is the well-known PLGA since this material improves drug release by making it more sustained and controlled. Although PLGA is generally considered to be biocompatible, in some cases, degradation of the products can cause adverse effects or localized toxicity. It is important to perform toxicity studies and evaluate potential side effects associated with PLGA degradation in each specific application. Furthermore, during its degradation, it can acidify the surrounding environment due to the release of lactic acid and glycolic acid, which can negatively affect certain tissues or cells, especially in applications where a neutral physiological environment is required. It would be interesting to explore other options, such as natural materials, which can improve the mechanical properties and, in some cases, provide antibacterial and anti-inflammatory properties without affecting the biocompatibility of the nanoparticles.

After reviewing the mentioned studies, characterization standards for PHBV nanoparticles need to be established in order to better compare studies between authors. In addition, researchers should give more detailed information when reporting the methods used, the results obtained, and the limitations of their studies, in order to be able to compare the data obtained in different laboratories around the world. Collaboration between scientists from different disciplines, such as chemistry, biology, medicine, and engineering, would be beneficial in addressing challenges from different perspectives. Also, many of the release experiments were focused on a short period of time. Larger studies would be beneficial to assess long-term stability and toxicity. At the same time, more in vivo studies are needed to better understand the effects of nanoparticles in more complex biological systems.

The studies mentioned in this paragraph are shown in a schematic form in Table 1**.**

### 2.2. PHBV Composites for Tissue Engineering Applications

Tissue engineering is a discipline of biomedicine that aims to promote the regeneration of the structure and functionality of damaged tissues in the body by using biomaterials capable of guiding metabolic processes and cellular activity. In this field, PHAs have been widely used for the synthesis of three-dimensional cellular scaffolds with improved structural and functional properties for bone and cartilage regeneration [60]. They can also be used as controlled-release systems of bioactive molecules and cellular signals to induce tissue regeneration. A specific example is the development of PLGA and PHBV nanoparticles incorporated into scaffolds to have a controlled release system of BMP-2 and BMP-7 morphogenetic proteins [61]. Currently, numerous studies have focused on optimizing the porosity and connectivity of these matrices with the finality of achieving proper integration of the biomaterial into the tissue, allowing its replacement by the cellular matrix as the polymer degrades, without compromising its mechanical properties or stability [62]. One of the most innovative and promising techniques for the synthesis of polymeric matrices with controlled porosity and connectivity is 3D printing. This method allows the growth of human mesenchymal stem cells inside the biopolymer and makes the space suitable for their cellular differentiation [63].

Another method to create biodegradable PHBV scaffolds is using selective laser sintering (SLS). It is demonstrated that the unique microstructure formed by this technique has the potential for conducting in vitro and in vivo tests [64]. In another publication, the degradation behavior of the scaffolds was investigated by incubating them in phosphate-buffered saline (PBS) for 6 weeks. SEM was used to characterize the microstructure of the scaffolds before and after incubation in PBS. The integrity of the structure was not affected and the molecular weight slightly decreased. The PHBV scaffolds fabricated with SLS generally showed adequate mechanical properties and good structural integrity after incubation [65]. The same technique was used by another group, in this case, to fabricate a nanocomposite of CaP/PHBV. The in vitro studies revealed a high viability of SaOS-2 cells and normal morphology and phenotype after 3 and 7 days of cultures on all scaffolds. The release behavior of bovine serum albumin (BSA) in this nanostructure was also studied. An initial burst release of BSA was observed, followed by a slow liberation of the protein. The PHBV matrix exhibited a slight degradation after 28 days of the experiment. It was concluded that nanocomposite scaffolds produced by SLS could be used in conjunction with biomolecules to enhance their functionality and apply them in bone-tissue engineering applications [66,67].

In another study that also aimed bone-tissue regeneration, the scaffolds were modified by introducing polyacrylamide (PAM) to the inner surface of the nanostructure by in situ ultra-violet polymerization. The reason for making this modification was to improve the biological and mechanical properties of PHBV scaffolds. The modified PHBV platforms had interconnected pores with porosities ranging from 75.4% to 78.6% and with a maximum volume of 20 to 50 microns. Sheep bone mesenchymal stem cells (BMSCs) were cultured on the pure PHBV scaffold and the modified ones. Hydrophilic PAM chains did not affect the BMSC viability or proliferation rate, but the initial cell adhesion (one hour) of the culture was significantly enhanced [68]. In another study, they tried to improve the properties by modifying the surface of the biomaterial by grafting dextran (DEX) onto the fibrous scaffolds of PHBV. DEX modification resulted in significantly improved surface hydrophilicity and bioactivity compared to unmodified scaffolds. The study found that the PHBV-DEX shells significantly promoted the growth of BMSCs compared to the unmodified ones [69]. New polymer mixtures of poly-L-lactic acid, PCL, and PHBV have also been explored. This polymer mixture was enriched with nanohydroxyapatite (nanoHA) and strontium-substituted nanoHA (SR-Nano-HA) to enhance osteogenic potential. Investigators tested the scaffolds in pre-osteoblastic cell culture and found no cytotoxic effects for 14 days. Composite platforms with nano-HA and SR-Nano-HA showed significantly higher levels of alkaline phosphatase (ALP) activity and calcium production compared to non-enriched ones. This indicates that composite scaffolds have greater osteogenic potential [70]. In another article, two types of polymers, PHB and PHBV, were mixed in different proportions using a salt-leaching technique. SEM was used to examine the porous surface of the scaffolds, FTIR to study the chemical interaction between the polymers, and water contact angle measurements were used to determine the hydrophilicity of the scaffolds. A hemolysis assay was also performed where the results showed that the mixed scaffolds were hemocompatible. Finally, different cell lines (Vero, Hela, and MDBK) were grown on the porous mats of the biopolymer scaffolds to test their viability and adhesion over time. The results showed that the mixed platforms had better cell viability, proliferation, and adhesion compared to the individual polymers [71]. Using a mixture of two polymers, PHBV and PLGA, it was found that increasing the amount of PLGA in the starting solution resulted in a larger pore size, higher wettability, and better thermal stability of the scaffolds. Researchers performed in vitro biological experiments to test the suitability of the platforms to support the colonization and differentiation of murine pre-osteoblastic cells towards an osteoblastic phenotype. They found that scaffolds with a higher PLGA content had higher proliferation rates [72]. Another study focused on a new mixture of the biopolymers PCL and PHBV containing hydroxyapatite (HA) nanoparticles included the use of the double-leaching technique. Morphology, porosity, degradation rate, FTIR spectra, and the surface and mechanical properties of the scaffold, as well as its cell attachment and proliferation capacity, were evaluated. Increasing the PHBV and HA content to 3 and 5% by weight in the PCL matrix caused an increase in porosity in all samples. Cell-culture experiments demonstrated that the PCL/PHBV/HA nanocomposite scaffold had a better predisposition for cell proliferation than the pure PCL/PHBV one. The use of PCL/PHBV/HA scaffolds has promising potential for bone-tissue engineering [73]. Using an electrospinning technique, another research group fabricated a PHBV scaffold with HA particles to mimic bone tissue and enhance the regeneration process. Morphology, fiber diameters, and scaffold composition were studied by SEM. Focused ion-beam sectioning was used to verify the integration of HA particles into PHBV fibers. The biological study was carried out on the MG-63 cell line for 7 days; also, a cell proliferation assay (CellTiter-Blue R© assay) was made. Cellular integration with the scaffold was visualized by confocal imaging and SEM. In this study, the modified scaffold improved early collagen and mineral deposit formation on osteoblasts after 7 days of incubation. The formation of these collagen fibrils on the surface of osteoblasts followed by mineralization is a typical sign of cell development towards bone tissue regeneration. [74]. A novel mixture of PHBV added minced fish scales into wet electrospun freeze-dried PHBV nanofibers. The morphology obtained was analyzed by SEM, and the human osteosarcoma MG-63 cell line was used to assess the biocompatibility of the composite scaffolds by measuring cell viability, ALP, and type I collagen production. The results indicated that the scale-modified scaffold has enhanced bioactivity and is useful in repairing bone-tissue damage [75]. In another study, a new multifunctional 3D fibrous scaffold was fabricated by coelectrospinning using a combination of hydrophobic PHBV/PCL fibers and hydrophilic pullulan (PUL) strands embedded in diatom shells (DS). This platform was used for loading and releasing a hydrophobic antibiotic, cefuroxime axetil. The in vitro studies were made with the human primary sarcoma cell line (Saos-2). The results suggested that the incorporation of DS into scaffolds may be a promising technique to enhance bioactivity in bone-tissue engineering [76]. In another publication, the objective of the study was to know the capacity of human-induced pluripotent stem cells (iPSCs) for bone differentiation on an electrospun PHBV-nanofiber scaffold. The results concluded that they were biocompatible and favored the proliferation of human iPSCs. Also, osteogenic differentiation was significantly improved in this scaffold compared to the tissue-culture polystyrene control. Moreover, the expression of two important osteogenic protein markers, osteocalcin, and osteopontin, was significantly overexpressed in the iPSC-loaded platforms compared to the control [77]. Another group of researchers wanted to compare the osteogenic abilities of HA and the osteogenic inducing medium (OIM). They used PHBV electrospun nanofibrous scaffolds and observed cell morphology, detected osteogenic markers, measured osteogenesis-related genes, and analyzed microRNAs. The studies found that HA was just as capable as OIM at differentiating mesenchymal stem cells (MSCs) into bone cells but at a slower rate. They also discovered that numerous microRNAs participate in the osteogenic differentiation of MSCs through different pathways [78].

The PHBV biopolymer has also been used in neural-tissue regeneration, where it promotes cellular adhesion, proliferation, and differentiation. Its potential in treating various neural injuries and diseases has been demonstrated [79]. With this aim, an electrospinning method was used to produce a chitosan-crosslinked nanofibrous biodegradable PHBV scaffold. The structure and cell culture assays using Schwann cells were microscopically, physically, and mechanically analyzed. The cells were able to grow fine on the created platform, indicating that this mixture of materials can be a promising candidate for applications in nerve conduits. The obtained results suggest that the generated material possesses suitable characteristics for further in vivo studies [80]. In another publication, the morphological and chemical properties of nanofibrous scaffolds composed of PHBV, PLA, and collagen were analyzed. Astrocyte differentiation and gene expression were measured by immunofluorescence and real-time–quantitative polymerase chain reaction (q-PCR) assays. The in vivo model was performed using a spinal cord injury (SCI) model in Sprague Dawley rats. The study demonstrated that PHBV/PLA/collagen scaffolds (70:30 and 50:50) could suppress glial fibrillary acidic protein expression in astrocytes. This molecule positively regulates the cells, which after an SCI, can become overactivated and form a scar that inhibits the regeneration of the damaged tissue. Therefore, the scaffold could decrease the accumulation of astrocytes in the injured area and thus create a more favorable environment for the infiltration of other cells. This enabled the protection of residual neurons and improved locomotor function in rats with the PHBV/PLA/collagen scaffolds. Overall, the study demonstrated the potential of these nanostructures for the treatment of SCI by regulating astrocyte activation and supporting neuronal survival [81].

There is another article that also discusses PHBV/PLA/collagen membranes for duroplasty after decompression in rats with SCI. The study focused on evaluating the material and the biological characteristics, subcutaneous implantation tests, and contusion SCI tests in rats to investigate the effects of the membranes on inflammasome activation and macrophage polarization. The results demonstrated that duroplasty with PHBV/PLA/Collagen membranes reduced glial scar formation and promoted axonal growth by inhibiting inflammasome activation and modulating macrophage polarization in acute SCI. Functional locomotor recovery improved 8 weeks after the injury [82].

Another research group conducted a study on PHBV nanofibers loaded with metallic silver particles to combat bacterial infections in joint arthroplasty. In vitro tests were carried out to evaluate the antibacterial activity of the nanofibers against *Staphylococcus aureus* and *Klebsiella pneumoniae*, as well as their compatibility with fibroblasts and osteoblasts. The results showed that only silver-containing PHBV nanofibrous scaffolds had high antibacterial activity and inhibited the growth of both bacteria. Furthermore, they found that nanofibrous scaffolds with less than 1% silver nanoparticles were free of cytotoxicity In vitro. Overall, the study suggests that PHBV nanofibrous scaffolds with this percentage of silver nanoparticles have the potential to be used in joint arthroplasty [83].

For repairing anterior cruciate ligament ruptures, some investigators developed a PHBV scaffold with high content of 3HV. The paper studied the biosynthesis of a variety of PHBV biopolymers with different 3HV contents. The investigation carried out the characterization of the platforms produced and biocompatibility tests were made on L929 fibroblasts. The results showed good cellular viability. They concluded that the aligned fibers network promotes fibroblast alignment in the axial fiber direction, which is desirable for ACL repair applications [84].

Various research groups are dedicated to exploring the most suitable materials and techniques for cartilage tissue engineering. One group of investigators used a conjugation of PHBV with type I collagen to produce a mechanically stable, biodegradable, and adhesive cell scaffold. The characterization of the scaffold was carried out using techniques such as SEM, Attenuated total reflection FTIR, atomic force microscopy, and electron spectroscopy for chemical analysis. Furthermore, the degradation and behavior of fibroblasts on nanofibrous scaffolds were studied. According to the article, electrospun PHBV/collagen composite nanofibrous scaffolds exhibited good mechanical properties, biocompatibility, and biodegradability. These scaffolds favored the growth and proliferation of fibroblast cells, and the cells demonstrated adequate adhesion and dispersion on the scaffolds. Results also revealed that the rate of degradation of the scaffolds could be controlled by adjusting the ratio of PHBV to collagen. This study suggested that electrospun PHBV/collagen composite nanofibrous scaffolds have potential applications in tissue engineering [85]. In a publication in 2022, one group of investigators studied the influence of the addition of Bioglass into PHBV porous platforms. Cartilage progenitor cells (CPCs) were seeded into the control and the PHBV/10% Bioglass scaffolds. The CPC-constructs were exposed to a 6-week in vitro chondrogenic induction culture and then transplanted in vivo for another 6 weeks to see the difference between the CPC-PHBV and CPC-PHBV/10% Bioglass platforms. Compared to the control, the PHBV/ 10% Bioglass scaffold had better properties and results like hydrophilicity, a higher percentage of adherent cells, and significant production of cartilage-like tissues. Also, the polymerase chain reaction analysis showed that aggrecan, collagen II, and SOX-9 from the CPC-PHBV/10% Bioglass scaffolds were more expressed compared to the CPC-PHBV ones. All indicate that the addition of Bioglass to PHBV can improve the chondrogenic differentiation of CPCs [86]. For the same application, researchers created a scaffold made of PHBV modified with a compound called quercetin (QUE) using a two-step surface modification method, resulting in a bioactive cartilage tissue-engineered scaffold called PHBV- g-QUE. The scaffold facilitates chondrocyte growth and maintains the chondrogenic phenotype resulting from the upregulation of aggrecan, collagen II, and SOX-9. In vivo assay performed in nude mice for 4 weeks showed that implantation of the PHBV-g-QUE fibrous scaffolds significantly promoted cartilage regeneration compared to PHBV by itself [87]. In another publication, they compared the chondrogenic capacities of BMSCs, CPCs, and chondrocytes. The different cell types were combined with a PHBV scaffold and cultured in vitro for one week without inducing chondrogenic differentiation. The created constructs were then transplanted subcutaneously into nude mice for six weeks. The results showed that CPCs could form cartilage better than BMSCs, and even outperform chondrocytes in this task [88]. Other researchers used the electrospinning technique to create a barium titanate (BaTiO3) scaffold with PHBV that mimics the structure of natural cartilage. The 20% BaTiO3 scaffold showed improved mechanical properties and a piezoelectric coefficient like native tissue. The in vitro studies show that polarized scaffolds highly promote cell attachment, proliferation, and collagen II gene expression against control (pure PHBV) and unpolarized scaffolds. The article concludes that the created piezoelectric scaffold is a good alternative for regenerating or repairing cartilage [89].

For wound healing, one group created a dressing made of a bilayer scaffold produced by electrospinning a hydrophilic PUL fibrous membrane (barrier layer) onto a wet electrospun hydrophobic PHBV fibrous mat (regenerative layer). The study optimized the production of PHBV using a bacterial strain called *Cupriavidus necator* and characterized the resulting polymer using various techniques, including Proton nuclear magnetic resonance (^1^H-NMR) and FTIR. The valerate molar percentage and the average molecular weight of the polymer were also determined using the ^1^H-NMR and SLS techniques, respectively. In vitro studies showed that the PUL membrane maintained L929 cell proliferation and prevented cells from migrating within the barrier phase, while the PHBV layer supported cell viability, proliferation, and migration, creating a regenerative 3D structure. The results showed that the new PUL/PHBV bilayer scaffold was a promising candidate for wound-healing applications [90]. Other researchers investigated the effects of different processing variables and the inclusion of PEG on the fibers of PHBV. They used FTIR and SEM to study the structure. The average diameter obtained ranged from 400 to 3000 nanometers, depending on the amount of PEG used. Cell growth was studied by performing fluorescence cell imaging experiments. They found that the cutaneous fibroblasts BJ-1 showed different adhesions and growth rates on the different types of PHBV fibers. The result indicated that PHBV mats with a low amount of PEG have a high potential for use in soft-tissue engineering, particularly in wound-healing applications [91]. In another publication, the authors designed an electrospun core-sheath mat made of polydioxanone (PDX) and PHBV. The purpose was to imitate the piezoelectric properties of biological tissues to stimulate cell growth and tissue regeneration. The article studied the fabrication of mixed biopolymer fibers in different weight ratios for scar-free wound healing. The physicochemical properties of the blend scaffolds were characterized using various techniques such as SEM, FTIR, X-ray diffraction, and differential scanning calorimetry. In vitro studies were performed to assess the biocompatibility of the scaffolds using fibroblasts, keratinocytes, and endothelial cells. In vivo studies were performed in Wistar rats to study the efficacy of scaffolds for promoting scar-free wound healing. The paper related the right balance of physicochemical and piezoelectric properties to the 20/80 PDX/PHBV composition. [92].

To fix abdominal wall defects, which can be caused by abdominal trauma or congenital rupture, silk fibroin and PHBV scaffolds were prepared. Characterization and cytotoxicity assays, in vitro tests to contemplate cell morphology and viability, and q-PCR to detect the gene expression of growth factor TGF-β1 and collagen I, were performed. In vivo studies were also achieved in Sprague Dawley rats. After 7 and 15 days of implanting the scaffold, an evaluation of their in vivo tissue regeneration capacity was performed. All the results pointed out that the hybrid nanofiber SF/PHBV scaffolds had a high efficiency and biocompatibility to repair abdominal wall defects [93].

As in the case of nanoparticles, some authors have explored the use of blends with other polymers, such as PLGA, PCL, and PLA. By incorporating these additional polymers, the scaffolds can exhibit improved strength, flexibility, and stability.

Natural materials such as collagen, chitosan, silk fibroin, and hyaluronic acid have also been incorporated into PHBV scaffolds. These natural materials have several advantages. For example, collagen is a major extracellular matrix component and provides structural support and bioactivity for cell attachment and tissue regeneration. Chitosan has excellent biocompatibility and can promote wound healing. Silk fibroin has biodegradability and mechanical properties that make it suitable for tissue-engineering applications. Hyaluronic acid, with its viscoelastic properties, is involved in tissue hydration and cell migration. The incorporation of these natural compounds can create a more favorable healing environment, enhance cellular interactions and improve tissue integration within the scaffold.

In the context of bone-tissue engineering, inorganic and ceramic materials such as hydroxyapatite, tricalcium phosphate, and bioactive glass have also been incorporated into PHBV scaffolds. These materials provide bioactivity and osteoconductive properties that promote bone regeneration. The addition of these inorganic components improves the mechanical properties of the scaffold and enhances its ability to support the growth and differentiation of bone cells.

Like in the case of nanoparticles, long-term studies are needed to assess the long-term effects and the immune responses due to the scaffold integration. These studies will show the true potential of the scaffold and its impact on the host organism. In addition, the authors should discuss regulatory aspects and the scalability of the developed systems to facilitate the implementation of these approaches to medicine.

In conclusion, despite the experimental progress made in the field, there is still a lack of standardization and a comprehensive understanding of the studies made.

The studies mentioned in this paragraph are shown in Table 2 in a schematic form.

## 3. Conclusions

In conclusion, biopolymers of the PHA family are very promising in a wide range of applications due to their natural origin and their excellent properties, including biocompatibility and biodegradability. In the field of medicine, these biomaterials are particularly relevant as their degradation products do not produce toxic agents for the organism. This biocompatibility is essential for their use in biomedical applications such as tissue engineering, implants, sutures, and controlled drug-delivery systems.

However, some of the physical and mechanical characteristics of some of the biopolymers may limit their use in specific applications. These limitations can be solved thanks to the great capacity of PHAs to modulate their properties by modulating different factors, like the selection of the microorganism to produce the biopolymer, the manipulation of the culture conditions, the composition of the biomaterial, and the biopolymer blend.

PHAs also degrade under normal environmental conditions in a relatively short time because they can be biodegraded by microorganisms, which significantly reduces their environmental impact. In contrast, conventional polymers can accumulate for long periods in ecosystems, causing pollution problems. This feature makes PHAs a promising option for addressing the problem of plastic waste and contamination.

In the specific case of PHBV, the variation in the percentage of hydroxyvalerate in the composition of the biomaterial makes a clear difference in its characteristics. The increase in this monomer leads to an increase in the flexibility and strength of the biopolymer, as well as making the particle more amorphous, which facilitates the diffusion of the encapsulated drug and has an impact on how drug release occurs. In a study mentioned in the article where the biopolymer was used with a 3% concentration of hydroxyvalerate encapsulating insulin, a hydrophilic drug, an initial-burst release was observed followed by a gradual release pattern of 63.2% after 27 days. Of this amount, 19% was released at a relatively fast rate within 24 h, while the rest was released gradually in a stationary phase. In another study where the produced nanoparticles had 12% hydroxyvalerate and encapsulated quercetin, most of the release occurred in the first 5 h of water immersion. At 15 days (360h) the signal obtained was equivalent to that of the empty nanoparticles, suggesting complete release. It should be noted that the amount of drug released was low as this is related to the low water solubility of the compound.

The other most used method to modulate the characteristics of the biomaterial was the use of biopolymer blends. Studies have shown that the incorporation of PLGA in composite materials such as PHBV/PLGA results in a higher encapsulation efficiency in hydrophilic substances. This is because PLGA acts as an additional hydrophobic barrier, preventing the rapid release of the hydrophilic substances and allowing for a more controlled and prolonged release. In some cases, they also use a second barrier to further enhance the controlled release. In an article mentioned in this review, they used hyaluronic acid/jeffamine hydrogels with the PHBV/PLGA polymer blend. The results showed that 63% of the teriparatide was released in 50 days in a more sustained and controlled manner than the nanoparticles and hydrogel alone.

In tissue engineering, in addition to the PHBV/PLGA mixture, other mixtures with collagen and calcium phosphate are also used. Collagen is used because it is a protein naturally present in tissues that form part of the extracellular matrix. The mixture of PHBV with collagen combines the structural properties of PHBV with the ability of collagen to promote cell adhesion, migration, and extracellular matrix synthesis. This mixture is especially useful in tissue-regeneration applications where enhanced bioactivity is sought. Moreover, calcium phosphate is an essential component of bone structure. Mixing PHBV with calcium-phosphate materials, such as hydroxyapatite, improves the mechanical properties of the scaffold and promotes bone mineralization and regeneration. This combination is particularly suitable for applications in bone regeneration and the repair of bone-tissue defects.

The remarkable flexibility of polyhydroxyalkanoates (PHAs) and the blends that can be made with them allows the creation of a wide variety of biopolymers with unique properties, making them a highly suitable choice for diverse applications in multiple industrial sectors, including biomedicine. In turn, these polymers exhibit significant advantages over conventional polymers, most notably, their ability to be environmentally friendly. This environmentally favorable characteristic increases their attractiveness and bodes well for their future.

Nowadays, there is still room for research and development in this field despite the wide range of existing publications on PHAs. New polymer blends can be explored and the properties of existing PHAs can be varied to obtain materials with improved properties to make them suitable for specific applications. Moreover, investigations can be focused on the development of more sustainable and economical production processes to facilitate the implementation of PHAs at the industrial level. This would allow biopolymers to compete more effectively with conventional materials in the marketplace.

## 4. Future Perspectives

PHAs have demonstrated their great potential in numerous studies, especially in the fields of tissue engineering and drug delivery. However, their use in clinical practice is not yet a reality, as there are still bottlenecks to overcome. Currently, the main obstacles related to biopolymers focus on the following research areas.

First, developing PHAs with new or improved physical properties. One example of research is the development of smart biomaterials, capable of responding to external signals such as changes in pH, temperature or light. This will allow for more advanced and sophisticated biomedical devices, such as drug-delivery systems triggered by specific signals.

The other crucial aspect that requires attention is the industrial production of these biomaterials. For biopolymers to be used in clinical practice, it will be key that existing methods no longer present challenges in terms of efficiency and cost. Studies in this line of research aim to optimize fermentation processes to increase productivity. They also aim to improve extraction and purification processes for better yields. Another key point is the raw material; although the use of whey as a carbon source to produce PHAs has already been mentioned, there are other cheap sources that should be explored in the future. These include derivatives of cellulosic compounds and other agro-industrial effluents. It should be noted that biocompatibility and safety standards for clinical application are high and must be rigorously met.

In summary, as research and technology advance, new applications and possibilities for PHAs are expected to emerge. The development of smart materials, improvements in industrial-scale production and compliance with stringent biocompatibility standards are and will be key areas of focus in PHA research.

## Figures and Tables

**Figure 1 ijms-24-11674-f001:**
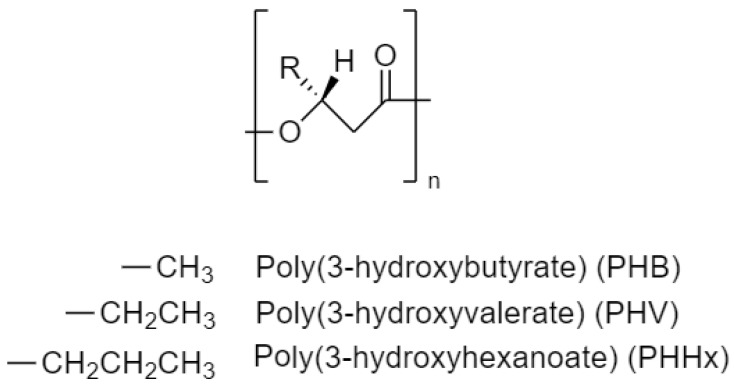
General structure of PHAs and some monomeric units of different polymers.

**Figure 2 ijms-24-11674-f002:**
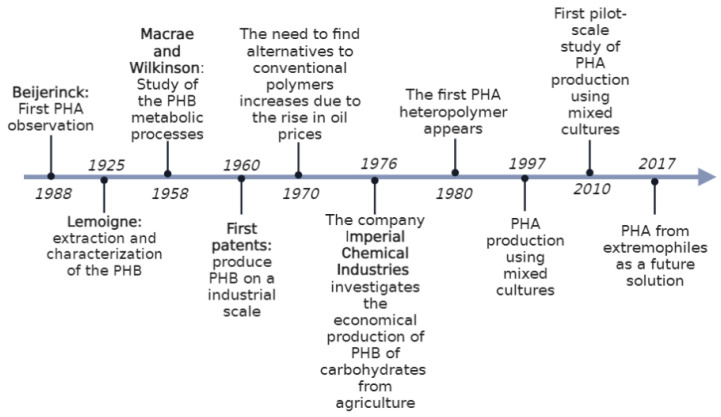
Historical timeline of PHAs. Modified from [13].

**Figure 3 ijms-24-11674-f003:**
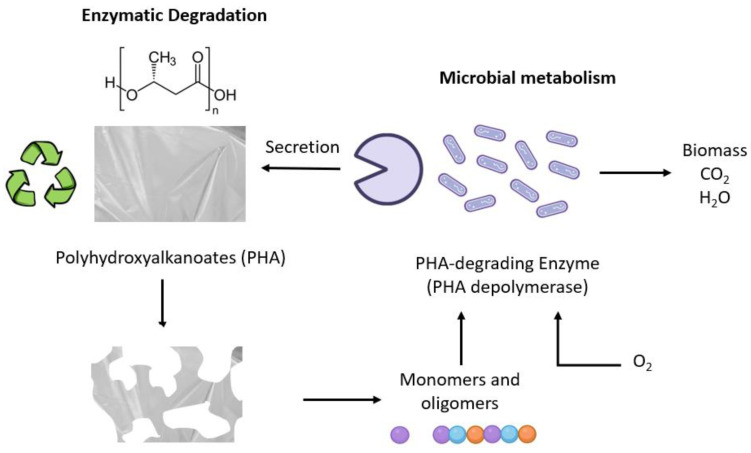
Biodegradation of PHAs in a natural environment. Modified from [17].

**Figure 4 ijms-24-11674-f004:**
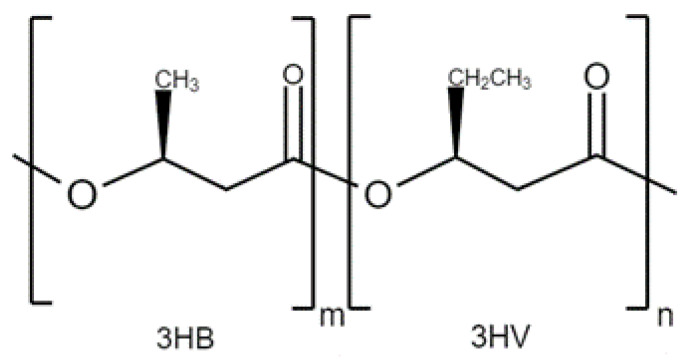
The general structure of PHBV.

**Figure 5 ijms-24-11674-f005:**
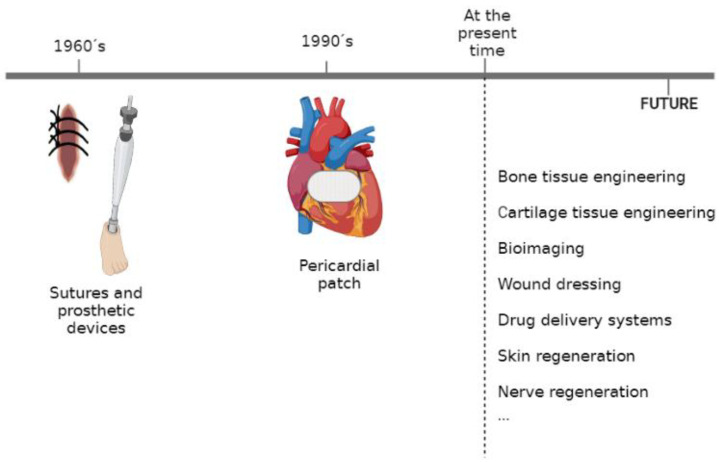
Applications of PHBV. Modified from [18].

**Figure 6 ijms-24-11674-f006:**
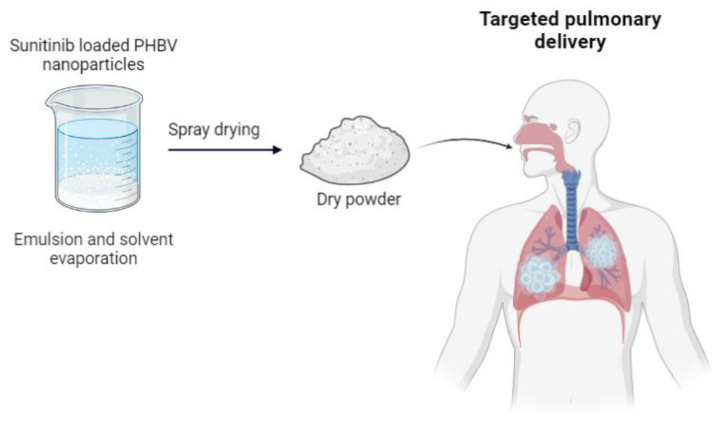
Sunitinib-loaded PHBV nanoparticles for targeted pulmonary drug delivery. Modified from [44].

**Figure 7 ijms-24-11674-f007:**
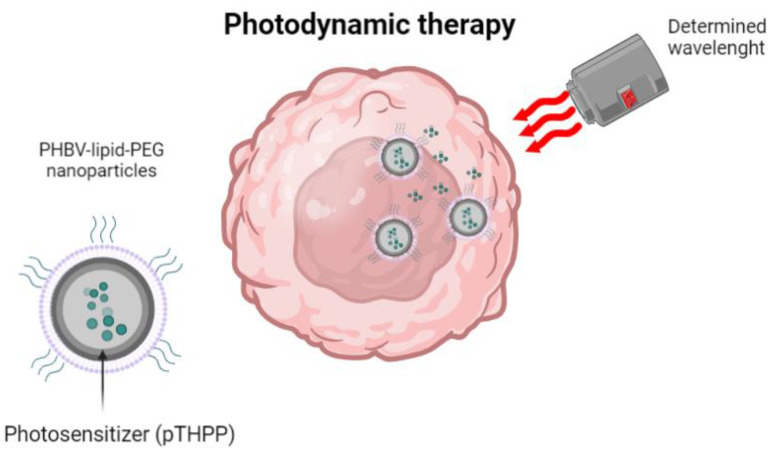
PHBV-lipid-PEG nanoparticles for photodynamic therapy. Modified from [46].

**Table 1 ijms-24-11674-t001:** Articles related to the use of PHBV as a drug-delivery system.

Nanoparticle	Method of Fabrication	Applications	Encapsulated Drug	Studies Performed	Ref.
PHBV	Emulsion and solvent evaporation method	Liver cancer, Hepatocellular carcinoma	Paclitaxel	Size and morphology Entrapment and drug loading efficient Fourier-transform infra-red spectroscopy X-ray diffraction Differential scanning calorimetry Thermogravimetry analysis Hemolysis experiment In vitro paclitaxel release Cell culture and cellular viability Cell uptake Xenograft tumor model In vivo antitumor efficacy In vivo biodistribution study Statistical analysis	[33]
PHBV/PLGA	Double-emulsion method	Colon cancer	5-Fluorouracil	Experimental design Determination of encapsulation efficiency Particle size and morphological studies Differential scanning calorimetry Thermogravimetry analysis Fourier-transform infra-red spectroscopy In vitro release study In vitro cellular uptake of nanoparticles Cytotoxicity assay Hemolysis assay In vivo anti-tumor activity Statistical analysis	[34]
PHBV/PLGA	Double-emulsion method	Colon cancer	5-Fluorouracil and oxaliplatin	Determination of encapsulation efficiency Particle size and morphological studies Differential scanning calorimetry In vitro release study In vitro cellular uptake of nanoparticles Cell viability Reactive oxygen species detection Apoptosis study Hemolysis analysis In vivo anti-tumor activity Statistical analysis	[35]
PHBV	Emulsion-diffusion method	Colon cancer	5-Fluorouracil	Determination of encapsulation efficiency Particle size and morphological studies In vitro release study Cytotoxicity analysis	[36]
PHBV	Nanoprecipitation	Colon cancer	Silymarin	Silymarin drug release Characterization of the nanocarriers by scanning electronic microscopy and atomic force microscopy In vitro cytotoxicity assessment of the Nanocarriers Statistical analysis	[37]
PHBV	Modified emulsion and solvent evaporation method	Breast cancer	Docetaxel	Colloidal morphology Particle size and surface charge Entrapment efficiency Analysis of drug by high-performance liquid chromatography In vitro drug release In vivo-in silico and in vitro–in vivo correlation studies Cytotoxicity Cell uptake Anticancer activity Drug–plasma interaction and hemolysis studies	[38]
PHBV	Emulsion and solvent evaporation method	Breast cancer	Docetaxel	Box–Behnken optimization studies Morphological studies Particle size and surface charge Entrapment efficiency In vitro drug release Pharmacokinetics study In vitro and in vivo correlation and in silico simulation Cell-line studies In vivo anticancer activity	[39]
PHBV	Modified double-emulsion and solvent evaporation method	Epithelial cell lines: -Ovarian cancer cells (SKOV-3) -Cervical cancer cells (HeLa)	NA	Dynamic light scattering Transmission electron microscopy Cell culture Flow cytometry Immunofluorescence Reducible biotin assay Western blot MTT assay	[40]
PHBV	Double-emulsion and solvent evaporation method	-Ovarian cancer cells -Endometrial cancer cells	Paclitaxel	Characterization of the nanoparticles Low-voltage electron microscopy Drug encapsulation efficiency Efficiency and release kinetics Ultra-performance liquid chromatography Fourier-transform infra-red spectroscopy All-atom molecular dynamics method Coarse-grain molecular dynamics methods Confocal laser scanning microscopy MTT assay and flow cytometry analysis	[41]
PHBV	Emulsion and solvent evaporation method	Lung carcinoma epithelial cells A549	Ellipticine	Particle size and zeta potential Scanning electron microscopy Drug encapsulation efficiency In vitro experiment Statistical analysis	[42]
PHBV	Oil-in-water emulsion technique	Lung carcinoma epithelial cells A549	Ellipticine	Characterization of the nanoparticles Drug loading efficiency In vitro cytotoxicity Statistical analysis	[43]
PHBV	Emulsion and solvent evaporation	Lung carcinoma epithelial cells A549	Sunitinib	Particle size, polydispersity index, and zeta potential Encapsulation efficiency and drug loading Drug-release study Study kinetics and mechanism of drug release Fourier-transform infra-red spectroscopy In vitro cellular uptake study In vitro cytotoxicity Scanning electron microscopy	[44]
Folic acid-functionalized PHB	Emulsion and solvent evaporation method	Osteosarcoma	Etoposide	Characterization of PHBV nanoparticles Determination of entrapment efficiency In vitro etoposide release studies Cytotoxicity assay Apoptosis/necrosis assay	[45]
PHBV-lipid-PEG	Nanoprecipitation technique combined with self-assembly	Photodynamic therapy	5,10,15,20-Tetrakis (4-hydroxy-phenyl)-21H,23H-porphine	Particle size analysis and zeta potential Measurement Determination of drug loading and entrapment efficiency X-ray Diffraction analysis Photophysical properties Cell culture experiments Intracellular uptake Photodynamic therapy studies Statistical analysis	[46]
Super-paramagnetic iron oxide functionalized PHBV	Double-emulsion and solvent evaporation method	Contrast medium Hyperthermia applications Tissue-specific drug delivery	Ceftiofur	Transmission electron microscopy Dynamic light scattering Ultraviolet-visible spectrophotometry Fourier-transform infra-red spectroscopy Differential scanning calorimetry Ceftiofur entrapment efficiency Cell cultures Antibacterial activity Vibrating sample magnetometer Cytotoxicity assays for polymeric nanoparticles Statistical analysis	[47]
PHBV	Double-emulsion and solvent evaporation method	Contrast medium for magnetic resonance imaging agent for hyperthermia Nanocarriers for targeted drug delivery	Super-paramagnetic iron oxide	X-ray diffraction of magnetite Analysis by electronic transmission microscopy Size and zeta potential of magnetite-loaded nanoparticles Magnetic measurements Magnetic accumulation of the loaded nanoparticles in intestinal tissue	[48]
PHBV/PLGA	Double-emulsion and solvent evaporation method	Osteoporosis	Teriparatide	Experimental design Particle size and size distribution Morphological studies High-performance liquid chromatography analysis Entrapment and drug-loading efficiency Fourier-transform infra-red spectroscopy X-ray powder diffraction Differential scanning calorimetry Intrinsic fluorescence spectroscopy In vitro teriparatide release Teriparatide stability in release medium Mathematical modeling	[49]
PHBV/PLGA loaded in hyaluronic acid/jeffamine hydrogel	Crosslinking reaction	Osteoporosis	Teriparatide	Characterization of the hydrogel Degree of swelling ratio Measurement of crosslinking efficiency Rheological characterizations Morphological characterizations of Teriparatide loaded delivery system In vitro teriparatide release studies Mathematical modeling of release data Stability of teriparatide Cytotoxicity studies In vivo studies	[50]
PHBV	Nanoprecipitation	Antibacterial properties	Epirubicin	Scanning electron microscope Energy dispersive X-ray spectroscopy Particle size analysis X-ray diffraction analysis Fourier-transform infra-red spectroscopy Drug-loading and encapsulation efficiency In vitro release studies Antibacterial studies	[51]
PHBV	Nanoprecipitation	Antibacterial properties	Different plant leaves	Characterization of guava extract: Thin layer chromatography, Fourier -transform infra-red spectroscopy, wide-angle X-ray diffraction analysis Characterization of nanoparticles Scanning electron microscopy Antibacterial studies Statistical analysis	[52]
PHBV	Double-emulsion method with high-speed homogenization	Antimicrobial, antimicrobial, anti-inflammatory, and antiviral properties	Quercetin	Dynamic light scattering Scanning electron microscopy Transmission electron microscopy X-ray diffraction Differential scanning calorimetry Thermogravimetry analysis The release profile of quercetin-loaded nanoparticles Cell culture and cytotoxicity assay In vitro release studies Analysis of insulin release kinetic	[53]
PHBV	Single- and double-emulsification and solvent evaporation technique	Immuno-suppression	Fingolimod	Experimental design study Characterization of fingolimod nanoparticles Size measurement, determination of loading efficacy, and capacity of the nanoparticles Determination of the morphology of the nanoparticles Stability study In vitro drug release study Evaluating release kinetics and the mechanism using a mathematical model	[54]
PHBV	Double-emulsification and solvent evaporation method	Diabetes	Insulin	Experimental design study Particle size, polydispersity index, and zeta potential measurement Morphological studies Determination of insulin encapsulation efficiency Fourier-transform infra-red spectroscopy X-ray diffraction Differential scanning calorimetry study Circular dichroism spectrophotometry	[55]
PHBV	Emulsion and solvent evaporation technique	Topical ocular administration	Hydrocortisone	Characterization of nanoparticles Entrapment efficiency and drug loading Cumulative release assay Evaluation of the ophthalmic toxicity Confocal studies	[56]
PHBV	Double-emulsion and solvent evaporation method	Parkinson’s disease	Pramipexole	Experimental design Particle size and size distribution Residual solvent determinations Morphological study High-performance liquid chromatography analysis Linearity and range Detection and quantification limits Encapsulation and drug-loading efficiency calculations Fourier-transform infra-red spectroscopy Differential scanning calorimetry In vitro pramipexole release profile Mathematical modeling	[57]
PHBV	Oil-in-water technique	Treatment of skin conditions	Retinyl palmitate and Dead Sea Water or MgCl_2_	Particle topography and size distribution analyses by scanning electron microscopy Retinyl palmitate loading and encapsulation efficiency In situ release of Dead Sea water and MgCl_2_ from micro/nanocapsules Determination of the hemolytic activity of the micro/nanoparticles Cell cultures Cell viability Determination of cytokines Production of reactive oxygen species Genotoxicity Detection of particle uptake by the cells Studies on penetration through human skin Statistical analysis	[58]
PHBV	Emulsion and solvent evaporation	Transdermal drug delivery systems	Fluorescent dye Nile Red	Nanoparticle topography with scanning electron microscopy Particle size distribution analysis In vivo studies Determination of the polymer of the nanocapsules in the skin with gas chromatography-mass spectrometry Histological preparation studies Statistical analysis	[59]

**Table 2 ijms-24-11674-t002:** Articles related to the use of PHBV for tissue engineering.

Scaffold	Method of Fabrication	Applications	Structural Modifications/Drug Loading	Studies Performed	Ref.
PHBV	Selective Laser Sintering	Bone-tissue engineering	NA	Microstructural characterization Relative density estimation Mechanical testing	[64]
PHBV	Selective Laser Sintering	Bone-tissue engineering	NA	In vitro degradation test Characterization	[65]
Ca-P/PHBV	Selective Laser Sintering	Bone-tissue engineering	NA	Characterization of microspheres and scaffolds Cell viability Cell proliferation Alkaline phosphatase activity Cell morphology Statistical analysis	[66]
Ca-P/PHBV	Selective Laser Sintering	Bone-tissue engineering	Bovine Serum Albumin	Characterization of bovine serum albumin-loaded microspheres and scaffolds In vitro release from microspheres and scaffolds Statistical analysis	[67]
PHBV and PHBV modify by the introduction of polyacrylamide	Particulate-leaching method modified on the inner surface of scaffolds using in situ Ultra-Violet polymerization	Tissue engineering: Sheep bone mesenchymal stem cells	NA	Chemical structure Porosity Pore size distribution Morphology Compressive properties Wettability Cell studies Statistical analysis	[68]
PHBV	Electrospun PHBV fibrous scaffolds that had been treated with methacrylic acid under ultra-violet light	Tissue engineering: Sheep bone mesenchymal stem cells	Dextran covalently attached to the surface of electrospun PHBV fibrous scaffolds	Characterization of tested samples Cell morphology investigation Statistical analysis	[69]
PLLA/PCL/PHB	Blend filaments with twin-screw extrusion Three-dimensional scaffolds were printed by fused deposition modelling	Bone-tissue engineering	Nano-hydroxyapatite and strontium-substituted nano-HA	Scaffold characterization In vitro biological evaluation of pre-osteoblasts in 3D composite scaffolds Adhesion and morphology of pre-osteoblasts within scaffolds Cell viability assessment within scaffolds Alkaline phosphatase activity measurement Calcium production Collagen production Statistical analysis	[70]
PHB/PHBV	Salt leaching technique	Tissue engineering applications	NA	Morphology and surface area analysis Fourier-transform infra-red spectrophotometer analysis Differential scanning calorimetry analysis Water contact angle Hemolysis assay In vitro cell viability assay In vitro cell attachment study Statistical analysis	[71]
PHBV/PLGA	Solution-extrusion additive manufacturing technique	Bone-tissue engineering	NA	Morphological characterization Acetone leaching and proton nuclear magnetic resonance analysis Contact angle measurements Thermal characterization Mechanical characterization Biological characterization	[72]
PCL/ PHBV	Dual-leaching technique	Bone-tissue engineering	Hydroxyapatite nanoparticles	Morphological observation of prepared scaffolds Porosity measurement Degradation rate Mechanical property Fourier-transform infra-red analysis Contact angle characterization In vitro biological study Cell viability assay Cell morphology analysis Data analysis	[73]
PHBV	Electrospinning	Bone-tissue engineering	Hydroxyapatite	Scaffold characterization Cell viability Cell imaging: confocal and scanning electron microscopy Statistical analysis	[74]
Fish scale/PHBV	Wet-electrospun and freeze-drying	Bone-tissue engineering	NA	Characterization of the fish scales Biocompatibility of the decellularized fish scales Characterization of fish scale/PHBV scaffolds Biomineralization studies In vitro studies of fish scale/PHBV scaffolds Histologic analysis Statistical analysis	[75]
Diatom shells and PHBV/PCL	Co-electrospinning system	Bone-tissue engineering	Cefuroxime axetil	Purification and characterization of diatomic shells Characterization of scaffolds Drug loading and release In vitro cell culture studies Statistical analysis	[76]
PHBV	Electrospun	Bone-tissue engineering	NA	Scanning electron microscopy MTT assay Alkaline phosphates activity Calcium content assay Real-time polymerase chain reaction Western blot Statistical analysis	[77]
PHBV	Electrospinning	Bone-tissue engineering	Hydroxyapatite	Fabrication and characterization of electrospun nanofibers Mesenchymal imaging Staining and semi-quantification of osteogenic markers Real-time quantitative polymerase chain reaction for gene and microRNA analyses Statistical analysis	[78]
PHBV	Electrospinning	Neural tissue regeneration	NA	LITERATURE REVIEW	[79]
PHBV	Electrospinning	Neural tissue regeneration	Cross-linked chitosan by chemical method	Structural characterization Cellular culture studies	[80]
PHBV/PLA/Collagen	Co-electrospinning	Neural tissue regeneration	NA	Characterization of nanofibrous scaffolds Cell cultures studies Scaffold biocompatibility and scanning electron microscopy evaluation Immunocytochemistry and Changes in the gene expressions of astrocytes In vivo experiments Hematoxylin and eosin and Immunofluorescence staining Western blotting analyses Behavioral testing Statistical analysis	[81]
PHBV/PLA/Collagen	Co-electrospinning	Neural tissue regeneration	NA	Characterization of nanofibrous substitutes Scaffold biocompatibility and Scanning electron microscopy In vivo experiments Hematoxylin and eosin and immunofluorescence staining Stereological assessment of spinal cord lesions Western blotting analyses Statistical analysis	[82]
PHBV	Electrospinning	In vitro antibacterial activity	Metallic silver particles	In vitro biodegradation Antibacterial assessment Silver release Cell adhesion Cell proliferation and viability Alkaline phosphatase activity	[83]
PHBV	Electrospinning	Anterior cruciate ligament	NA	Nuclear magnetic resonance Surface property characterization Characterization of thermal properties Wide-angle X-ray diffraction Mechanical testing Cytotoxicity testing Cell morphology observation Statistical analysis	[84]
PHBV/collagen	Electrospinning	Cartilage tissue engineering	NA	Surface characterization In vitro biodegradation Cell-counting assay	[85]
PHBV/ 10% Bioglass	Solvent casting/ particulate leaching method	Cartilage tissue engineering	NA	Properties of the PHBV and PHBV/10% Bioglass scaffolds Hydrophilicity, Water absorption, and cell-adhesion determination Cell proliferation Chondrogenic induction in vitro Characterization of in vivo tissue-engineered cartilages Quantitative real-time quantitative polymerase chain reaction Statistical Analysis	[86]
PHBV-g-QUE	Two-step surface modification method	Cartilage tissue engineering	NA	Characterization of PHBV fibrous scaffolds and PHBV-g-QUE fibrous scaffolds Cell culture studies with PHBV fibrous scaffolds and PHBV-g-QUE fibrous scaffolds Cartilage regeneration evaluation. In vivo Statistical analysis	[87]
PHBV	Solvent casting particulate	Cartilage tissue engineering	NA	Cell proliferation in vitro Wet weight and volume measurement Glycosaminoglycan and total collagen Histology and immunohistochemistry GAG, total collagen, and biomechanical analysis Real-time quantitative polymerase chain reaction Enzyme-linked immunosorbent assay Statistical analysis	[88]
PHBV and Barium Titanate (BaTiO3)	Electrospinning	Cartilage Regeneration	NA	Characterization of the nanofiber scaffolds Piezoelectric coefficient In vitro cell culture study Statistical Analysis	[89]
Pullulan-PHBV	Wet and dry electrospinning	Wound healing	NA	Hydrogen nuclear magnetic resonance analysis of the polymer Calculation of valerate mole percentage in the copolymer Determination of average molecular weight of PHBV by static light scattering analysis Fourier transform infra-red spectroscopy analysis of PHBV Differential scanning calorimetry analysis Fiber morphology analyses of the scaffold by scanning electron microscopy Enzymatic degradation and water retention study Tensile strength analyses Bacterial transmission, oxygen, and water vapor permeability analyses of scaffolds In vitro cell culture study Statistical analyses	[90]
PHBV/PEG	Electrospinning	Wound healing	NA	Morphology of electrospun mats Chemical analysis Water uptake Enzymatic degradation Cells and incubation conditions Statistical analysis	[91]
Piezoelectric core–shell PHBV/PDX	Electrospinning	Wound healing	NA	Characterization of fibrous PDX/PHBV mats Dehydration of scaffolds for Scanning electron microscopy MTT assay Analysis of the RAW 264.7 cell morphology Enzyme-linked immunosorbent assay TNF-α Fibroblast spheroid formation and cell migration assay Fibroblast-induced contraction of scaffolds In vivo biocompatibility tests Wound-healing studies Histological analysis Epithelial length from the wound edges Calculation of wound contraction indices Calculation of granulation tissue scoring Statistical analysis	[92]
Silk fibroin/PHBV	Electrospinning	Abdominal wall	NA	Characterizations of the electrospun nanofibers In vitro cytocompatibility In vivo biocompatibility Statistical analysis	[93]

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
