# Peer review of "Biomedical Applications of the Biopolymer Poly(3-hydroxybutyrate-co-3-hydroxyvalerate) (PHBV): Drug Encapsulation and Scaffold Fabrication"

_ijms, 2023, doi:10.3390/ijms241411674_

Round 1
Reviewer 1 Report
Herein, the authors present a fairly comprehensive review of the Biomedical applications of PHBV. The only shortcomings I see are: 1) section 5. Conclusions and future perspectives. This section should be split into two; one for conclusions and one for future perspectives: the conclusions are vague/non-specific and the future perspectives mostly include creating polymer blends; the authors should discuss more deeply the direct of the field and where specific problems arise. The other shortcoming would be that more Figures and diagrams should be included in regards to the various applications to better compare and contrast the previous work.
Reviewer 2 Report
The review by S.M.D. Prado et al. considers the achievements in the biomedical applications of PHBV biopolymers and functional composites on their basis as drug delivery systems and tissue healing materials. This review theme is a “hot spot”, however the authors should indicate clearly the specific emphasis of this review in comparison with other review on this topic. So, a position of this review among relevant review papers should be highlighted and latest achievement in this area should be analyzed.
This paper may be published after major revision by addressing the following issues.
1. The relevant paper The Bacterial Cellulose (BC) Biocomposites for Potential Use in Biomedical Applications. Polymers 2022, 14, 5544. https://doi.org/10.3390/polym14245544 should be cited in the paper.
2. The sufficient part of this review is the general information about PHBV, which can be find elsewhere, for instance in review Polymers 2018, 10, 732; doi:10.3390/polym10070732. This information fits more a tutorial, than a critical review.
3. A critical overview of the results placed in the Tables 1,2 should be done in the paper. The references on these Tables should be done in the text body.
4. The conclusion and future outlook section should be more informative. In this paper. They are too short.
5. The reference list should be formatted.
English Lanuage used in this paper is readable and quite understandable.
Round 2
Reviewer 2 Report
The authors hae corrected properly this manuscript accoreding to referee's comments. it may be published in the present form.
English in this paper is quite understandable.